# Outbreak of a Systemic Form of Camelpox in a Dromedary Herd (*Camelus dromedarius*) in the United Arab Emirates

**DOI:** 10.3390/v13101940

**Published:** 2021-09-28

**Authors:** Sunitha Joseph, Joerg Kinne, Péter Nagy, Jutka Juhász, Rajib Barua, Nissy Annie Georgy Patteril, Donata Hoffmann, Florian Pfaff, Bernd Hoffmann, Ulrich Wernery

**Affiliations:** 1Central Veterinary Research Laboratory, Dubai 597, United Arab Emirates; sjoseph@cvrl.ae (S.J.); jkinne@cvrl.ae (J.K.); nissygeorgy@yahoo.com (N.A.G.P.); cvrl@cvrl.ae (U.W.); 2Farm and Veterinary Department, Emirates Industry for Camel Milk and Products, Dubai 294236, United Arab Emirates; peter@camelicious.ae (P.N.); jutkajuhasz@gmail.com (J.J.); dr.rajib@camelicious.ae (R.B.); 3Friedrich-Loeffler-Institut, Institute of Diagnostic Virology, Suedufer 10, 17493 Greifswald—Insel Riems, Germany; donata.hoffmann@fli.de (D.H.); Florian.Pfaff@fli.de (F.P.)

**Keywords:** camelpox, virus, systemic form, dromedary camel, next generation sequencing, phylogenetic analysis

## Abstract

Camelpox virus (CMLV) is the causative agent of camelpox, which frequently occurs in the Old World camelids-rearing countries except for Australia. It has also been described in experimentally inoculated New World camelids. Camelpox outbreaks are often experienced shortly after the rainy season, which occurs twice a year on the Arabian Peninsula because of the increased density of the insect population, particularly mosquitos. A systemic form of camelpox outbreak in seven dromedary camels was diagnosed by histology, virus isolation, and PCR. A phylogenetic analysis using full length CMLV genomes of the isolated CMLV strains showed a single phylogenetic unit without any distinctive differences between them. The United Arab Emirates (UAE) isolate sequences showed phylogenetical relatedness with CMLV isolates from Israel with only minor sequence differences. Although the sequences of viruses from both countries were closely related, the disease manifestation was vastly different. Our study shows that the virulence is not only determined by genetic features of CMLV alone but may also depend on other factors such as unknown aspects of the host (e.g., age, overall fitness), management, and the environment.

## 1. Introduction

Camelpox virus (CMLV) is the causative agent of camelpox, which frequently occurs in Old World camelids, the dromedary (*Camelus dromedarius*), and Bactrian camels (*C. bactrianus*). It has also been experimentally induced in New World camelids (NWCs) [1]. The infective agent of camelpox is taxonomically classified within the genus *Orthopoxvirus* of the family *Poxviridae*. The family is divided into two subfamilies: *Chordopoxviridae*, which infects vertebrates, and *Entomopoxviridae*, which are found in insects. Phylogenetic analyses of CMLV revealed that the virus is closely related to the Variola virus, the causative agent of smallpox in humans [2], which has been eradicated worldwide. Zoonotic transmission of CMLV to humans has been described several times with recent examples during an outbreak of camelpox in the northwest region of India and eastern Sudan [3,4]. Although being a zoonotic disease, camelpox is currently of limited public health importance, as cases of camelpox infection in humans are rare and only cause mild symptoms [5].

Camelpox is the most frequent infectious viral disease of the camel and therefore more widely reported. The disease occurs wherever camel husbandry is practiced [6]. An exception is the Australian dromedary population, in which camelpox has yet to be observed. Camelpox epidemics occur in regular cycles that are dependent upon the rainy season and the relationship of the density of the insect population to the number of immune camels in the population [7].

In camelids, CMLV causes proliferative dermatitis, which primarily affects younger animals [6]. Comparable pox-like lesions in camelids may also be induced by specific parapoxviruses causing camel contagious ecthyma, papillomaviruses, and other orthopoxviruses [8]. However, infection with orthopoxviruses other than CMLV has so far only been described in NWCs which were severely infected with cowpox virus (CPXV) [9,10]. Both localized and generalized external pox lesions of the skin have been described. Internal pox lesions, however, are rare as poxviruses are generally epitheliotropic, and reports on them are very limited [5,11].

Here we report an outbreak of camelpox in a dromedary herd in the United Arab Emirates (UAE) in summer 2020 that caused severe external and internal lesions in several animals. In addition, the complete genomes of different isolates were molecularly characterized by next generation sequencing, and comprehensive phylogenetic analyses were performed.

## 2. Materials and Methods

### 2.1. Clinical Evaluation of Diseased Animals

Clinical data were collected at the premises of the Emirates Industry for Camel Milk and Products (EICMP), located in Dubai, United Arab Emirates (N25°, E55°). EICMP owns the largest dairy camel herd in the world, with a total dromedary population of over 7500 heads (at the time of this outbreak) that are kept at two separate units. More detailed data of the affected herd unit 1 was summarized in Table 1. A herd health management program is in place and under strict veterinary control that includes, among many other elements, the clinical evaluation of the health status of all animals twice a day. The first camelpox in an adult female dromedary was diagnosed on 23 May 2020, that was followed by several new cases in various age groups until the last clinical case was detected on 26 July 2020. Sick animals were treated with supportive therapy (antipyretic: Ketoprofen, 3 mg/kg; vitamins: C and B-complex, intravenous fluids) and broad-spectrum antibiotics (Ceftiofur, 1 mg/kg; Marbofloxacin, 4 mg/kg) daily. During the outbreak, several animals that died were sent to the Central Veterinary Research Laboratory (CVRL) for necropsy. Table 2. Shows a list of these animals: one adult male (Table 2: number 1), two adult females (Table 2: numbers 6 and 7), three yearlings > 2 years of age (Table 2: numbers 2,3, and 4), and one male calf < 1 year of age (Table 2: number 5). Table 2 also shows the clinical signs of the necropsied dromedaries that were included in this study. All carcasses were thoroughly inspected, and any alterations recorded. Samples were taken from multiple organs and tested for CMLV using routine diagnostic methods such as histopathology, polymerase chain reaction (PCR), and virus isolation. In addition to these animals, two other calves with external pox lesions died due to different causes (one died of peritonitis, and the other one died of lung abscesses caused by *Corynebacterium pseudotuberculosis,* which was isolated on blood agar), and therefore were not included here.

### 2.2. Histopathology and Polymerase Chain Reaction (PCR)

Histopathological analysis was performed on different organs after fixing them in 10% formalin. The sample was embedded in paraffin, sectioned, and stained with haematoxylin and eosin, according to standard protocols [12].

The PCR is a fast and feasible method for the diagnosis of camelpox [13]. DNA was extracted by using the DNeasy Tissue Kit (Qiagen, Hilden, Germany) according to the manufacturer’s instructions. Gel-based PCR was performed as described in the OIE terrestrial manual in a total reaction volume of 50 µL [5]. The PCR were performed on the ABI 7500 Dx real-time PCR instrument (Applied Biosystems, Waltham, Massachusetts, USA). The cycling conditions were 94 °C for 5 min, 29 cycles at 94 °C for 1 min, 45 °C for 1 min, 72 °C for 2.5 min. Following a final extension at 72 °C for 10 min an aliquot of 10 µL of PCR product was analyzed in a 1% agarose gel.

### 2.3. Virus Isolation

Samples suspicious for CMLV were propagated in Vero cells and on the chorioallantoic membrane (CAM) of embryonated chicken eggs. Vero cells (ATCC, CCL-81) were routinely maintained in Minimal Essential Medium (MEM, Gibco, Fisher Scientific, Loughborough, UK) supplemented with 10% fetal bovine serum (FBS, Gibco, Thermo Fisher Scientific, Darmstadt, Germany), 1% penicillin-streptomycin (Sigma Aldrich, Germany), and 0.3% Amphotericin B (Sigma Aldrich-Merck, Darmstadt, Germany). Scabs and tissue samples were triturated and a 10% (*w*/*v*) suspension in MEM was prepared. The suspension was clarified by centrifugation and supernatant was further diluted 1:10 in MEM. Vero cell monolayer grown in 75 cm^2^ flask was inoculated with 2 mL of diluted supernatant and incubated at 37 °C for 1 h. Infected cells were maintained using MEM at 37 °C for 5–7 days and were monitored daily for cytopathic effect (CPE).

To isolate the virus in eggs, lysates from tissue samples were layered on CAM of 10–11 day old specific pathogen-free embryonated chicken eggs using a 23 G sterile needle. Inoculated eggs were incubated at 37 °C for 5 days. The inoculated eggs were opened and the CAM was examined for the presence of characteristic pock lesions.

### 2.4. Sequencing and Phylogenetic Analysis

Viral DNA was extracted from infected Vero cells using MasterPure™ Complete DNA and RNA Purification Kit (Lucigen, Middleton, Wisconsin, USA) according to the manufacturer’s instructions. The genomic DNA was then submitted to a commercial company (Eurofins Genomics, Ebersberg, Germany) for library construction and shotgun sequencing using an Illumina NovoSeq running in 150 bp paired-end mode. The resulting reads were quality trimmed using Trim Galore (version 0.6.6) with automated adapter detection and FastQC (version 0.11.9; [14]) for visual inspection of general quality aspects. The quality trimmed reads were then used for de novo assembly using Unicycler (version 0.4.4; [15]) and the resulting contigs were screened for sequences matching that of CMLV sequences using diamond blastx (version 0.9.21; [16]). Matching contigs were arranged with respect to orthopoxvirus genomes and eventually missing or unclear regions were further confirmed using manual inspection of mapped reads. The full length CMLV genomes were annotated to CPXV reference genomes, as CPXV represent the most complete and well annotated members of the orthopoxviruses. For phylogenetic analysis, the complete sequences obtained from this study were aligned with publicly available CMLV complete sequences using MAFFT (version 7.450; [17]). Here, Taterapox virus was used as the outgroup. The accession numbers of the strains used are included into the figures. The alignment was further screened for gaps and non-conserved regions using gBlocks (version 0.91b; [18]). A maximum likelihood phylogenetic tree was then calculated using FastTree (version 2.1.11, model GTR+gamma; [19]) and visualized using Geneious Prime (version 2021.0.1).

## 3. Results

The number of different types of dromedaries affected by this camelpox outbreak at Unit 1 of the farm is presented in Table 1. The morbidity rate was 1.1%, as 56 animals of various age groups showed typical skin lesions and were clinically sick. Most of the camels affected (*n* = 45, 80.4%) were above two years of age, but 16.1% (*n* = 9) were calves between four and six months of age. The mortality rate for the entire population at Unit 1 was 0.1%, but 8.9% (*n* = 5 out of 56) of the clinically affected dromedaries died of generalized camelpox. The other two mortalities occurred at Unit 2 of the farm. Details of clinical signs of the seven camels that died are presented in Table 2 and the results of lab investigations are summarized in Table 3.

All affected dromedary camels had developed fever, nasal discharge, external and internal camelpox lesions, as shown in Figure 1A–D. All but one individual exhibited perianal/scrotal skin lesions, and three animals presented skin lesions all over the body. Interestingly, all dromedary camels displayed pox lesions in internal organs as well, despite the age of the individual. The analyses of tissues using classical histopathology techniques demonstrated characteristic cytoplasmic swelling and ballooning of keratinocytes (Figure 2A). Bronchitis or pneumonia was evident in three animals (Table 2, Figure 2B).

CMLV could be isolated from each of the seven necropsied dromedary camels using several tissue samples (Table 3). The supernatant of inoculated Vero cell culture with a positive typical cytopathic effect (CPE) was tested by PCR for the successful propagation of the CMLV. The virus was not isolated from the digestive tract or from the liver, spleen, kidney, pancreas, and brain. The isolated CMLVs produced plaques on Vero cells after 1–4 days incubation with foci of rounded cells, giant cell formation, syncytia, and cell detachment (Figure 3A). The CAM displayed typical characteristic pock lesions as dense, greyish-white pocks as shown in Figure 3B.

Using next-generation sequencing, the full genomes of five CMLV isolates from this study were sequenced (Genbank accession numbers MZ300856-MZ300860). The length of the genomes ranged between 201,277 and 201,406 bp for D1865/20 and D1734/20, respectively. Deviating genome lengths between the isolates were exclusively caused by differences in the assembly of the terminal repeats at both ends of the CMLV genome. Sequences of all five CMLV isolates were nearly identical, showing only minor differences in homo-polymeric or repetitive regions. 228 gene coding sequences (CDS) were annotated in all genomes. For D1795/20 and D1804/20, a four base-pair deletion was observed at the end of gene CPXV196, causing a frameshift and altered position of the stop codon. Furthermore, D1795/20 and D1804/20 encoded for CPXV200 which was truncated by a frameshift in all other isolates.

A phylogenetic analysis using the full length CMLV genomes showed that CMLV seems to be rather conserved, showing only very few differences among the different isolates. Isolates obtained during this outbreak represent a single phylogenetic unit without distinctive differences (Figure 4A). Their closest relative is a CMLV isolate (Negev2016) that was obtained from a case of camelpox in Israel in 2016 (99.9% nucleotide identity) [20]. Isolates from Kazakhstan and Iran are slightly more different from the isolates obtained in this study (99.8% nucleotide identity) and they seem to represent a distinct phylogenetic group. In order to include more sequence data into this analysis, we calculated a phylogenetic tree based on the CMLV hemagglutinin (HA) gene using 59 sequences in total (Figure 4B). The phylogenetic tree again showed two separated groups, and isolates obtained from this study showed the highest nucleotide identity to strains from Ethiopia, Syria, and Israel. A direct comparison between CMLV genomes from the outbreak in United Arab Emirates and their closest relative virus Negev2016 from Israel revealed only very minor sequence differences. In detail, the terminal repeats and the core region differ in 16 SNPs, of which only two resulted in an amino acid substitution. Other differences can be attributed to copy number variations in short repeats or homopolymers. A duplication event within the terminal repeats of Negev2016 results in a truncated CPXV0008 gene when compared to the CMLV strains from the outbreak in the United Arab Emirates.

## 4. Discussion

Camelpox is the most important viral disease of camels and occurs in almost every country in which camel husbandry is practiced. Transmission of the virus is by direct contact between infected and susceptible camels, but transmission by arthropod vectors play a greater role, as outbreaks often arise during or after the rainy season [7]. An increased density of the tick *Hyalomma dromedarii* during the rainy season may also be responsible for the spread of the disease [7]. The morbidity rate of camelpox is variable and depends on whether the virus is circulating in a herd. The mortality rate is higher in young animals than in adults [21], as also experienced with the current outbreak. This is true especially when camels suffer from internal camelpox, as multiple pox-like lesions develop on the mucous membranes of the mouth, trachea, and lung. These lesions are a focus for secondary bacterial invaders. Pox lesions are also observed in the retina of the eye and cause blindness [6].

The findings of the present outbreak partly confirm previous reports but also provide some new observations on the importance, clinical manifestation and transmission of the disease. Camelpox has been observed previously at the premises of EICMP, but the occurrences have been sporadic, with only a few cases (*n* < 5) per year, and generalized pox resulting in death was extremely rare (one case per two to three years). Therefore, no earlier specific preventive measure (i.e., vaccination) was applied to control the disease. Compared to the previous years, this constitutes a relatively severe outbreak of camelpox at EICMP. However, even this outbreak affected only a small portion of the population, with a 1.1% morbidity rate and the overall mortality rate also quite low (0.1%). This outcome could be related to the good general condition and herd immunity of this dromedary population. In accordance with earlier observations, both mortality and morbidity rates were numerically higher in yearling animals above two years of age. However, the occurrence of the disease was not related to rainfall, as the outbreak started at the time of very high ambient temperatures (>40 °C) without any rain. Camelpox was first detected in a cohort of adult lactating females that had been imported from abroad two months earlier and were recently released from quarantine. Yearlings above two years of age were only affected several days later during the second wave of the outbreak following the movement of animals within the farm. Interestingly, calves of affected adult dromedaries that were kept together with their dams had not developed any clinical signs, with only one exception. In addition, two pregnant dromedaries had fever with pox-like skin lesions, but only one of them aborted six weeks later, while the other camel carried the pregnancy to term and gave birth to a healthy male calf. The virus was not isolated from the aborted fetus; hence, this abortion cannot be related to the CMLV infection.

It has been suggested that different strains of the CMLV may show some variation in their virulence, which may explain the severity of the disease in the dromedary camels under investigation [22], but the genetic base of this suggestion is poorly understood [23]. Therefore, publicly available different CMLV genome sequences have been compared to the sequences generated in this study. As full genome sequences of CMLV isolates or outbreak scenarios are limited, single-gene comparison was also performed. Clearly, all generated sequences of this studied outbreak were almost identical to each other. This fits with a disease outbreak that can be traced back to a single virus incursion. Interestingly, in phylogenetic analyses of both the full-length genome and the partial HA sequence, the isolates from this outbreak in the United Arab Emirates closely resembled sequences of a CMLV obtained from an outbreak in Israel. Even though both sequences are closely related, the clinical signs observed during both outbreaks, as well as the mortality rate, suggests a different degree of virulence for both viruses. During the outbreak in Israel in summer 2016, mainly female animals were affected and clinical signs included weakness, fever, abortions, and multifocal lesions of the skin. However, no animals died, and all completely recovered from the infection [20]. This contrasts with the severe clinical signs observed during this outbreak in the United Arab Emirates. This may indicate that the mortality is not only determined by the genetic features of CMLV, but may also depend on other factors, such as unknown aspects of the host (e.g., age, co-infections or overall fitness), management, and the environment. However, to calculate back from sequence information to virulence or mortality data is still not possible due to limited sequence data retained in public repositories. Recently, systemic infections with CMLV were also reported from India, where 15 dromedary camels showed massive clinical symptoms [24]. Only a 243 bp fragment of the conserved C18L gene was sequenced for virus confirmation. These limited sequence data are insufficient for the analysis of potential virulence markers.

There could be several reasons why this particular outbreak manifested in severe clinical signs and was also widespread across the premises of the farm as opposed to the earlier occurrence of camelpox when symptoms were milder and the disease was contained successfully. In this case, a large number of most likely susceptible dromedaries were introduced to a herd that had not encountered this strain of CMLV earlier. We assume that the general immunity of these animals was impaired due to major changes in the environment. This factor, together with the increased virulence of CMLV, could have contributed to the severe progression of the disease. In addition, the spread of camelpox within the farm was likely associated with herd management issues. These included the increased density of ticks (*H. dromedarii*) due to scanty ecto-parasite controls at the start of the COVID-19 pandemic. In our study, no investigation for CMLV in ticks was performed. Furthermore, unnecessary movement of both healthy and diseased camels and the lack of proper isolation of infected animals could be critical factors. We believe that these management issues were important predisposing factors contributing to this camelpox outbreak.

## 5. Conclusions

In conclusion, CMLV genomes generated from a camelpox outbreak of systemic disease in dromedary camels were almost identical to each other, and the closest relative was CMLV isolates from Israel. Although the sequences of viruses from both countries were closely related, the disease manifestation was vastly different, most probably due to different predisposing factors such as host susceptibility and herd management. Our study shows that the genetic features of CMLV alone are not the determining factor of its virulence. In this context, future studies should focus on genome analyses of CMLV strains isolated from localized and generalized forms of camelpox cases, as this will provide further insight into the factors affecting its virulence.

## Figures and Tables

**Figure 1 viruses-13-01940-f001:**
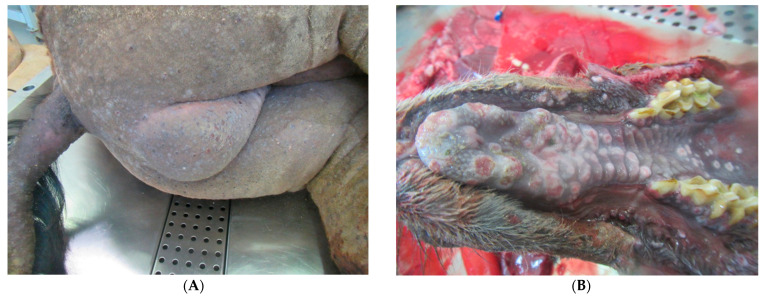
(**A**) Classical generalized camelpox lesions on the rear side of a camel as raised circular plaques (D1621/20). (**B**) Multiple pock erosions at the palate (D2053/20). (**C**) Pock lesions in the tracheal mucosa as raised circular nodules (D1621/20). (**D**) Pock nodules (yellow arrow) well circumcised, round and slightly raised in the lung parenchyma (D1795/20).

**Figure 2 viruses-13-01940-f002:**
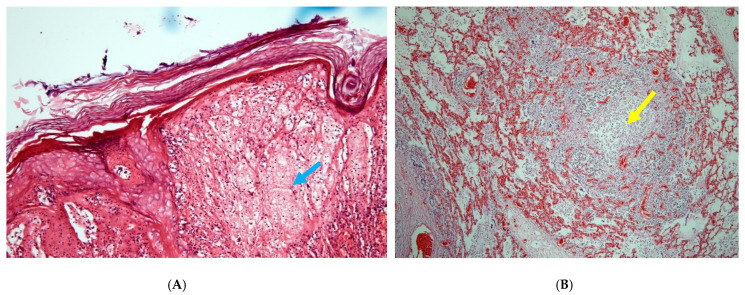
(**A**) Skin with focal marked cytoplasmic swelling and ballooning vacuolation (blue arrow) of keratinocytes in the stratum spinosum of the epidermis. HE, 120× magnification (D1804/20). (**B**) Lung with consolidated focus consisting of alveoles containing a mixture of infiltrating mononuclear cells (yellow arrow) as well as cytoplasmic and nuclear debris (HE, 120× magnification, D2053/20).

**Figure 3 viruses-13-01940-f003:**
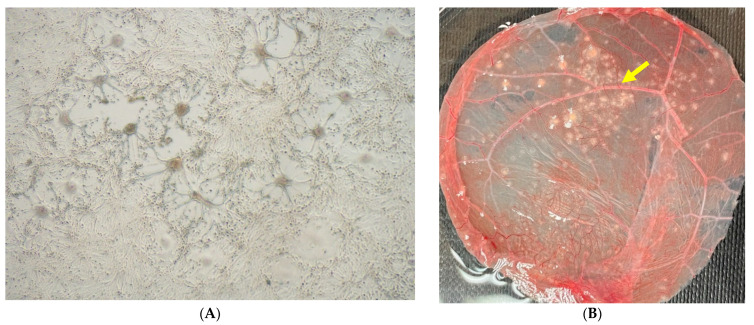
(**A**) Typical camelpox plaque on Vero cells after 72 h of incubation at 37 °C, 40× magnification (D1621/20). *(***B***)* Characteristic camelpox lesions on the CAM of embryonated chicken eggs (yellow arrow), 5 days after incubation at 37 °C 10× magnification (D1621/20).

**Figure 4 viruses-13-01940-f004:**
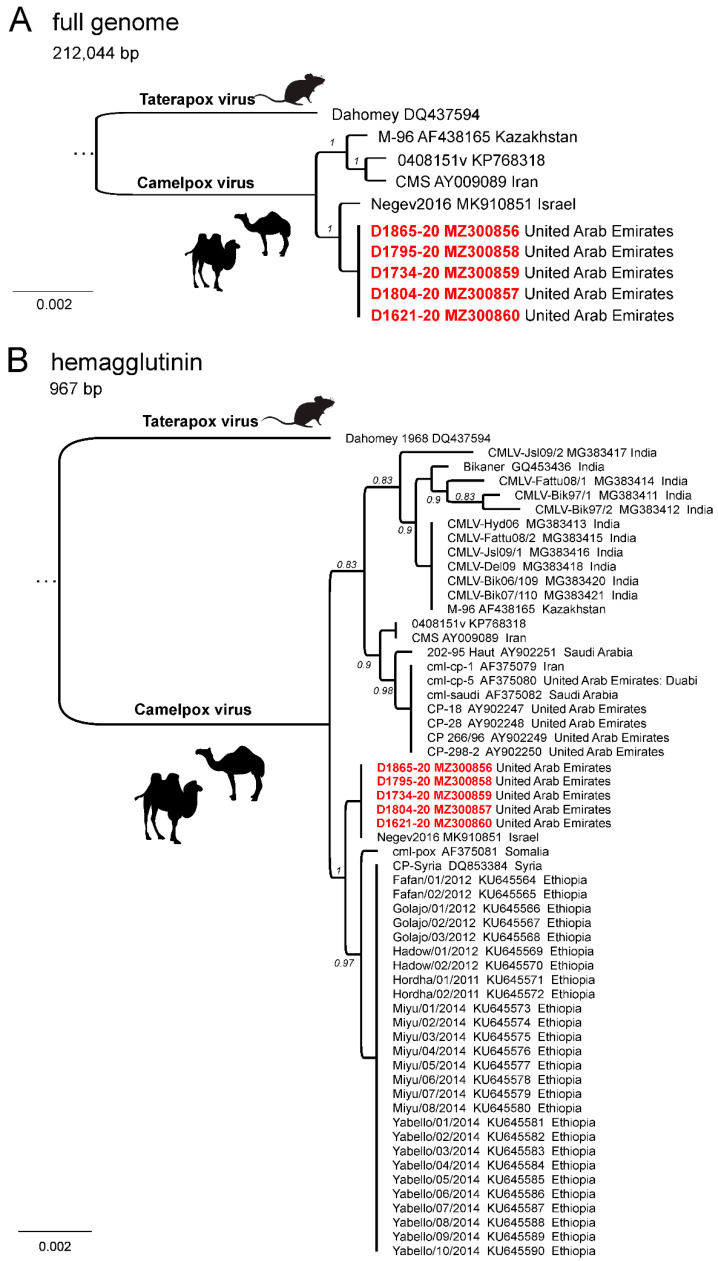
Maximum likelihood phylogeny based on the complete genome (**A**) and the partial hemagglutinin gene (**B**) sequences of Camelpox virus genomes from the outbreak in United Arab Emirates in comparison to available reference sequences. Taterapox virus is used as an outgroup.

**Table 1 viruses-13-01940-t001:** Number of different types of dromedaries affected by camelpox disease at unit 1 of EICMP.

Type of Dromedary	Number of Animals	Number of Diseased Animals	Morbidity Rate (%)	Number of Dead Animals	Mortality Rate (%)	Case Fatality Rate (%) *
Adult female	2689	24	0.89	2	0.07	8.3
Adult male	48	1	2.08	1	2.08	100.0
Calf < 1 year	1128	9	0.80	1	0.09	11.1
Yearling from 1 to 2 years	480	2	0.42	0	0.00	0.00
Yearling > 2 years	740	20	2.70	1	0.27	5.0
Total	**5085**	**56**	**1.10**	**5**	**0.10**	**8.9**

* Number of dead animals divided by the number of diseased animals.

**Table 2 viruses-13-01940-t002:** Clinical signs of dromedaries suffering from camelpox recorded seven days before their death.

No	Diagnostic ID and Type of Dromedary	Weight (kg)	Clinical Signs
1	D1621/20 Adult male	604	Fever, nasal discharge, general oedema but particularly on the preputium, off feed,
2	D1734/20 Yearling > 2 years	280	Fever, oedema around larynx/distal neck, Swollen body lymph nodes, off feed
3	D1795/20 Yearling > 2 years	211	Fever, nasal discharge, abdominal oedema, off feed
4	D1804/20 Yearling > 2 years	175	Fever, nasal discharge, many ticks, off feed
5	D2053/ 20 Calf male < 1 year	60	Fever, nasal discharge, poor body condition, not suckling
6	D1865/20 Adult female	456	Fever, general oedema, swollen body lymph nodes, off feed
7	D2132/20Adult female	310	Fever, nasal discharge, off feed

**Table 3 viruses-13-01940-t003:** Gross pathological and histopathological as well as virus isolation details of camelpox virus infection in a dromedary camel herd.

No	Diagnostic ID and Type of Dromedary	Lesions		Histology Lesions	Successful Virus Isolation From
		**External**	**Internal**		
1	D1621/20 Adult male	Legs, preputium, scrotum, nostril, lips	Gum, trachea, oesophagus, lung	Proliferative dermatitis, ballooning, pox-like inclusion bodies and pneumonia	Skin, body lymph nodes, trachea, oesophagus, gum and lung
2	D1734/20 Yearling female >2 years	Legs	Gum, trachea, oesophagus, lung	Proliferative dermatitis with bacterial infection	Skin, udder lymph nodes, lung, oesophagus
3	D1795/20 Yearling female >2 years	Pock nodules all over the body	Lung	Proliferative dermatitis, ballooning, bronchitis pox-like inclusion bodies, pneumonia	Skin, lung
4	D 1804/20 Yearling female >2 years	Pock nodules, especially in inguinal and perianal regions, swollen body lymph nodes, subcutaneous haemorrhages at the head	Lung	Proliferative dermatitis with pox-like inclusion bodies	Skin, body lymph nodes, trachea, tonsil, liver, spleen, brain, kidney, gum and lung
5	D2053/20 Calf male <1 year	Pock nodules all over the body	Mouth mucosa, gum, tongue, oesophagus, lung	Proliferative bronchitis, pox-like inclusion bodies	Skin, lips, nasal swabs, lung
6	D 1865/20 Adult female	Ventral abdomen/inguinal and perianal regions	Gum, trachea	Massive congestion and marked proliferation of follicular and parafollicular lymphatic tissue	Skin, gum, trachea, lung, liver
7	D2132/20 Adult female	Pock nodules all over the body	Lung, trachea	Proliferative dermatitis	Skin

## Data Availability

The data presented in this study are included in the manuscript or are available on request from the corresponding author. The virus genomes sequenced in this study have been deposited in GenBank using the accession numbers MZ300856–MZ300860.

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
