# Peer review of "Outbreak of a Systemic Form of Camelpox in a Dromedary Herd (Camelus dromedarius) in the United Arab Emirates"

_viruses, 2021, doi:10.3390/v13101940_

Round 1

Reviewer 1 Report

The Authors present an interesting study on an outbreak of camelpox in a dromedary herd. A systemic form was diagnosed in seven animals by histology, virus isolation and PCR and a phylogenetic analysis showed a single unit related with CMLV isolates from Israel. Anyway the disease manifestation was very different showing that CMLV virulence can be determined by genetic features, but also by factors related to host, management and environment.

The manuscript is well written and I believe only few clarifications are required to make the study complete.

I have the following few minor comments to further strengthen the manuscript. 

Table 1-2-3 -  it would be clearer to standardize the title of the first column (table 1) and second column (table 2 and 3): type of dromedary or gender or age classes?

p.7-8-9 –  it could be useful for the reader to add an arrow in each figure, mainly in figure 1D, 2A, 2B, 3B

p.12 line 24 – maybe an adjective is missing

What about serological investigations? Did you investigate on immunologic defenses of animals? If not, can you explain why?

Some minimal editing errors in References (country names with a lowercase initial).

Broadly, the study is well written, discussion is interesting, and the literature contains current contents.

Reviewer 2 Report

The paper write by Joseph et al. have a good quality but need some modifications. Their described and reported the isolation of multiple camelpoxviruses from an enlarge epidemic situation in  a cattle in Dubai.

The abstract section is too short and need to be re-write.

In the abstract section, authors could add some references about the cycle in camel pox and discuss more about that.

Please add the following paper: https://www.sciencedirect.com/science/article/pii/S0166354211004414

Please rephrase in the introduction “defined” and “used”.

Authors need to add elements of therapeutics inn the 2.1: which antibiotic and their doses? Vitamins and antipyretic….

For Corynecbaterium authors need to explain their material and method to identify it please.

2.2 “standard proctocol”?

Which percent of FBS?

Section 2.4 add references for all softwares or algorithms used.

Which software was used to visualize phylogenetic trees.

How many camel have documented corynebacteria infections? It’s not clear in the result section.

Sentence page 9 are out of context . please introduce length of all genomes and thei main properties. Before introduce the deletion.

Page 10=>  “99,9%” in whole genome analysis?

So phylogenetic analysis of whole genome could be name: Phylogenomic

What is the diferrence on the whole genomes between Israel and Dubai sequences please.

Ticks present on the animal there were not test for Camelpoxvirus? Author could precise this in the discussion section.

Virulence could be lead by host factors I agrre with authors.

Many viruses seems to be “dependant” for their virulences of host factors.

Round 2

Reviewer 2 Report

Many corrections have been addressed by authors. some points of discussion are not complete investigate (rainy season, vectors etc...)